# Institutional dashboards on clinical trial transparency for University Medical Centers: A case study

**Delwen L. Franzen** *, **Benjamin Gregory Carlisle**, **Maia Salholz-Hillel**, **Nico Riedel**, **Daniel Strech**

Berlin Institute of Health at Charité - Universitätsmedizin Berlin, QUEST Center for Responsible Research, Berlin, Germany

* delwen.franzen@bih-charite.de

**Data Availability Statement:** The authors confirm that all data underlying the findings are fully available without restriction. The dashboard is openly available at: https://quest-cttd.bihealth.org/. Code to produce the dashboard is openly available

## Abstract

### Background

University Medical Centers (UMCs) must do their part for clinical trial transparency by fostering practices such as prospective registration, timely results reporting, and open access. However, research institutions are often unaware of their performance on these practices. Baseline assessments of these practices would highlight where there is room for change and empower UMCs to support improvement. We performed a status quo analysis of established clinical trial registration and reporting practices at German UMCs and developed a dashboard to communicate these baseline assessments with UMC leadership and the wider research community.

### Methods and findings

We developed and applied a semiautomated approach to assess adherence to established transparency practices in a cohort of interventional trials and associated results publications. Trials were registered in ClinicalTrials.gov or the German Clinical Trials Register (DRKS), led by a German UMC, and reported as complete between 2009 and 2017. To assess adherence to transparency practices, we identified results publications associated to trials and applied automated methods at the level of registry data (e.g., prospective registration) and publications (e.g., open access). We also obtained summary results reporting rates of due trials registered in the EU Clinical Trials Register (EUCTR) and conducted at German UMCs from the EU Trials Tracker. We developed an interactive dashboard to display these results across all UMCs and at the level of single UMCs. Our study included and assessed 2,895 interventional trials led by 35 German UMCs. Across all UMCs, prospective registration increased from 33% ($n = 58/178$) to 75% ($n = 144/193$) for trials registered in ClinicalTrials.gov and from 0% ($n = 0/44$) to 79% ($n = 19/24$) for trials registered in DRKS over the period considered. Of trials with a results publication, 38% ($n = 714/1,895$) reported the trial registration number in the publication abstract. In turn, 58% ($n = 861/1,493$) of trials registered in ClinicalTrials.gov and 23% ($n = 111/474$) of trials registered in DRKS linked the

in GitHub at: https://github.com/quest-bih/clinical-dashboard. Code to generate the dataset displayed in the dashboard is openly available in GitHub: https://github.com/maia-sh/intovalue-data/releases/tag/v1.1. Data can be downloaded from the dashboard and are openly available on OSF at: https://osf.io/26dgx/. Raw data obtained from trial registries are openly available on Zenodo at: https://doi.org/10.5281/zenodo.7590083. Data for summary results reporting in the EUCTR are available via the EU Trials Tracker.

**Funding:** This work was funded by the Federal Ministry of Education and Research of Germany (BMBF 01PW18012, https://www.bmbf.de). The funders had no role in study design, data collection and analysis, decision to publish, or preparation of the manuscript.

**Competing interests:** I have read the journal's policy and the authors of this manuscript have the following competing interests: The authors are affiliated to the Charité – Universitätsmedizin Berlin, one of the institutions included in this evaluation and in the dashboard.

**Abbreviations:** CONSORT, Consolidated Standards of Reporting Trials; CTIMP, Clinical Trial of an Investigational Medicinal Product; DOI, Digital Object Identifier; DORA, Declaration on Research Assessment; DRKS, Deutsches Register Klinischer Studien (German Clinical Trials Register); EUCTR, EU Clinical Trials Register; FDAAA, Food and Drug Administration Amendments Act; ICMJE, International Committee of Medical Journal Editors; OA, Open Access; OSF, Open Science Framework; STROBE, Strengthening the Reporting of Observational Studies in Epidemiology; TRN, trial registration number; UMC, University Medical Center; WHO, World Health Organization.

publication in the registration. In contrast to recent increases in summary results reporting of drug trials in the EUCTR, 8% ($n = 191/2,253$) and 3% ($n = 20/642$) of due trials registered in ClinicalTrials.gov and DRKS, respectively, had summary results in the registry. Across trial completion years, timely results reporting (within 2 years of trial completion) as a manuscript publication or as summary results was 41% ($n = 1,198/2,892$). The proportion of openly accessible trial publications steadily increased from 42% ($n = 16/38$) to 74% ($n = 72/97$) over the period considered. A limitation of this study is that some of the methods used to assess the transparency practices in this dashboard rely on registry data being accurate and up-to-date.

## Conclusions

In this study, we observed that it is feasible to assess and inform individual UMCs on their performance on clinical trial transparency in a reproducible and publicly accessible way. Beyond helping institutions assess how they perform in relation to mandates or their institutional policy, the dashboard may inform interventions to increase the uptake of clinical transparency practices and serve to evaluate the impact of these interventions.

## Author summary

### Why was this study done?

- Clinical trials are the foundation of evidence-based medicine and should follow established guidelines for transparency: Their results should be available, findable, and accessible regardless of the outcome.

- Previous studies have shown that many clinical trials fall short of transparency guidelines, which distorts the medical evidence base, creates research waste, and undermines medical decision-making.

- University Medical Centers (UMCs) play an important role in increasing clinical trial transparency but are often unaware of their performance on these practices, making it difficult to drive improvement.

### What did the researchers do and find?

- We developed a pipeline to evaluate clinical trials across several established practices for clinical trial transparency and applied it in a cohort of 2,895 clinical trials led by German UMCs.

- We found that while some practices are gaining adherence (e.g., prospective registration in ClinicalTrials.gov increased from 33% to 75% over the period considered), there is much room for improvement (e.g., 41% of trials reported results within 2 years of trial completion).

- We developed a dashboard to communicate these transparency assessments to UMCs and support their efforts to improve.

### What do these findings mean?

- Our study demonstrates the feasibility of developing a dashboard to communicate adherence to established practices for clinical trial transparency.

- By highlighting areas for improvement, the dashboard provides actionable information to UMCs and empowers their efforts to improve.

- The dashboard may inform interventions to increase clinical trial transparency and be scaled to other countries and stakeholders, such as funders or clinical trial registries.

## Introduction

Valid medical decision-making depends on an evidence base composed of clinical trials that were prospectively registered and reported in an unbiased and timely manner. The registration of clinical trials in publicly accessible registries informs clinicians, patients, and other relevant stakeholders about what trials are planned, in progress or completed, and aggregates key information relating to those trials. Trial registration thus reduces bias in our understanding of the existing medical evidence and disincentivizes outcome-switching and selective reporting [1]. For clinical trials to generate useful and generalizable medical knowledge gain, trial results should also be reported in a timely manner after trial completion per the World Health Organization (WHO) Joint Statement on Public Disclosure of Results from Clinical Trials [2]. Disclosure is a necessary but not sufficient component of transparency: Trial results should also be openly accessible and findable, in line with established guidelines [2–6]. However, several studies have shown that clinical trials are often not registered and reported according to these standards [7–11].

Audits of research practices can build understanding of the status quo, inform new policies, and evaluate the impact of interventions to support improvement. Examples include the European Commission's Open Science monitor [12], the German Open Access monitor [13], the French Open Science Monitor in health [14], and institution-specific dashboards of select research practices [15]. Focusing on trial transparency, the EU Trials Tracker and the Food and Drug Administration Amendments Act 2007 (FDAAA) TrialsTracker [16,17] display up-to-date summary results reporting rates of public and private trial sponsors in a transparent and accessible way. The EU Trials Tracker served as a key resource for initiatives aiming to increase reporting rates of drug trials in the EU Clinical Trials Register (EUCTR) [18,19]. Based on the EU Trials Tracker, results reporting in the EUCTR has increased from 50% in 2018 to 84% (late 2022).

Research institutions such as University Medical Centers (UMCs) can incentivize practices for research transparency through their reward and promotion systems [20,21] and by providing education, infrastructure, and services [22,23]. However, internal and external assessments of research conducted at UMCs rarely acknowledge these practices [24,25]. Rather, traditional indicators of research performance such as the number of clinical trials, the extent of third-party funding, and the impact factor of published papers emphasize quantity over quality, which can entrench problematic research practices [26]. Initiatives such as the Declaration on Research Assessment (DORA) and the Hong Kong Principles have called for a change in the way researchers are assessed, and for more recognition of behaviors that strengthen research integrity [20,27]. The establishment of the Coalition on Advancing Research Assessment

(CoARA) and the 2022 Agreement on Reforming Research Assessment emphasize this shift towards rewarding responsible research practices to maximize research quality and impact [28]. In turn, the UNESCO Recommendation on Open Science adopted in 2021 affirmed the need to establish monitoring and evaluation mechanisms relating to open science [29]. Audits of transparency practices could empower UMCs to support their uptake by highlighting where there is room for improvement and where to allocate resources. Comparative assessments between institutions could also provide examples of successes and stimulate knowledge transfer.

Audits that are based on open and scalable methods facilitate repeated evaluation and uptake at other organizations. Such an evaluation of transparency practices at the level of clinical trials led by UMCs requires reproducible and efficient procedures for (a) sampling all clinical trials and associated results publications affiliated to UMCs and (b) measuring select registration and reporting practices. We previously established procedures for identifying all clinical trials associated with a specific UMC and their earliest results publications [9,11]. In turn, an increasing number of open-source publication and registry screening tools have been developed in the context of meta-research projects aiming to increase research transparency and reproducibility [10,30–32].

The objective of this study was to perform a status quo analysis of a set of established practices for clinical trial transparency at the level of UMCs and present these assessments in the form of an interactive dashboard to support efforts to improve performance. While the general approach of our study is applicable for UMCs worldwide, this study focused on German UMCs.

## Methods

Producing a dashboard for clinical trial transparency required the development of a pipeline consisting of 3 main steps: first, the identification of registered clinical trials led by German UMCs; second, the evaluation of select registration and reporting practices, including (a) the partly automated and partly manual identification of earliest results publications of these trials and (b) the application of automated tools at the registry and publication level; third, the presentation of these baseline assessments in the form of an interactive dashboard. An overview of the dependence of these steps on automated versus manual approaches is provided in **S1 Supplement**. The development of the dashboard was iterative and did not have a prospective protocol. The methods to develop the underlying dataset of clinical trials and associated results publications, however, were preregistered in Open Science Framework (OSF) for trials completed 2009 to 2013 [33] and 2014 to 2017 [34].

### Data sources and inclusion and exclusion criteria

The data displayed in the dashboard relate exclusively to registered (either prospectively or retrospectively) clinical trials obtained from 3 data sources with the following inclusion and exclusion criteria:

1. The IntoValue cohort of registered clinical trials and associated results [35]. This dataset consists of interventional clinical trials registered in ClinicalTrials.gov or DRKS, considered as complete between 2009 and 2017 per the registry, and led by a German UMC (i.e., led either as sponsor, responsible party, or as host of the principal investigator). Trials were searched for 38 German UMCs based on their inclusion as members on the website of the association of medical faculties of German universities [36] at the time of data collection. In line with WHO and International Committee of Medical Journal Editors (ICMJE) definitions [4,37], trials in this cohort include all interventional studies and are not limited to Clinical Trials of an Investigational Medicinal Product (CTIMP) regulated by the EU's

Clinical Trials Regulation or Germany's drug or medical device laws. The dataset includes data from partly automated and partly manual searches to identify the earliest reported results associated with these trials (as summary results in the registry and as publication). The methods for sampling UMC-specific sets of registered clinical trials and tracking associated results are described in detail elsewhere [9,11]. Briefly, we used automated methods to search registries for clinical trials associated with German UMCs and manually validated the affiliations of all trials. We deduplicated trials in this cohort that were cross-registered in ClinicalTrials.gov and DRKS (see more information in **S2 Supplement**). Results publications associated with these trials were identified by means of a manual search across several search engines. This was complemented by automated methods to identify linked publications in the registry [10]. To reflect the most up-to-date status of trials, we downloaded updated registry data for the trials in this cohort on 1 November 2022 and reapplied the original IntoValue exclusion criteria: study completion date before 2009 or after 2017, not considered as complete based on study status, and not interventional. More detailed information on the inclusion and exclusion criteria can be found in **S2 Supplement**.

2. For assessing prospective registration in ClinicalTrials.gov, we used a more recent cohort of interventional trials registered in ClinicalTrials.gov, started between 2006 and 2018, led by a German UMC, and considered as complete per study status in the registry. We downloaded updated registry data for the trials in this cohort on 1 November 2022 and reapplied the same exclusion criteria as above except for completion date (**S2 Supplement**).

3. For assessing results reporting in the EUCTR, we retrieved data from the EU Trials Tracker on 4 November 2022 [16]. We found a sponsor name for 34 of the UMCs included in this study as of August 2021 (sponsor names in the EU Trials Tracker are subject to change). If more than one corresponding sponsor name was found for a given UMC (Bochum, Giessen, Heidelberg, Kiel, Marburg, and Tübingen), we selected the sponsor with the most trials. More detailed information can be found in **S3 Supplement**.

## Analysis of registration and reporting practices

The dashboard displays the performance of UMCs on 7 recommended transparency practices for trial registration and reporting. In this study, we focused on adherence to ethical principles and reporting guidelines that apply to all trials. Compliance with a legal regulation was only assessed for summary results reporting in the EUCTR. For an overview of these practices, relevant guidelines and laws, the sample considered, and the measured outcome, see **Fig 1** (sources in **S4 Supplement**) and **Table 1**. The data for these metrics were obtained through a combination of automated approaches and manual searches, several of which have been described previously [8–11]. In the following, we outline the methods used to generate the data for each metric. More detailed information can be found in the Methods page of the dashboard and in **S5 Supplement**.

**Prospective registration.** Raw registry data downloaded from ClinicalTrials.gov and DRKS were further processed to determine the registration status of trials. We defined a trial to be prospectively registered if the trial was registered in the same or a previous month to the trial start date.

**Bidirectional links between registry entries and associated results publications.** We extracted links to publications from the registry data and obtained the full text of publications. We then applied regular expressions to detect publication identifiers in registrations, and trial registration numbers (TRNs) in publications. The application of these methods on the IntoValue cohort was reported previously [10].

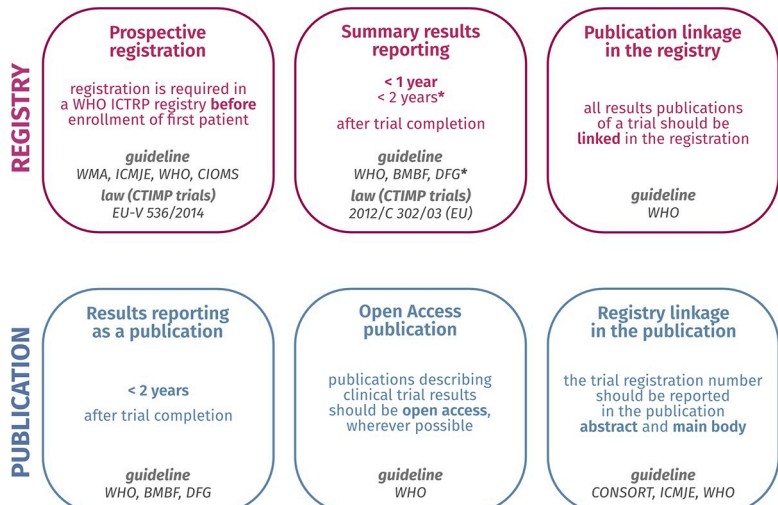

**Fig 1. Overview of the clinical trial transparency practices included in the dashboard.** Relevant guidelines and/or laws are provided for each practice (as of November 2022). A list of references can be found in **S4 Supplement**. An adaptation of this overview is included in the "Why these practices?" page of the dashboard. *DFG: According to the DFG guidelines at the time of writing, summary results should be posted in the registry at the latest 2 years after trial completion, or earlier if required by applicable legal regulations. BMBF, Bundesministerium für Bildung und Forschung; CIOMS, Council for International Organizations of Medical Sciences; CONSORT, Consolidated Standards of Reporting Trials; CTIMP, Clinical Trial of an Investigational Medicinal Product; DFG, Deutsche Forschungsgemeinschaft; ICMJE, International Committee of Medical Journal Editors; ICTRP, International Clinical Trials Registry Platform; WHO, World Health Organization; WMA, World Medical Association.

**Summary results reporting in the registry.** For ClinicalTrials.gov, we extracted the relevant information from the structured summary results field. For DRKS, we detected summary results based on the presence of keywords (e.g., Ergebnisbericht or Abschlussbericht) in the reference title. The summary results date in DRKS was extracted manually from the registry's change history. We obtained summary results reporting rates in the EUCTR from the EU Trials Tracker. We retrieved historical data (percent reported, total number of due trials, and total number of trials that reported results) from the associated code repository [16].

**Reporting as a manuscript publication.** The earliest publication found for each trial and its publication date was derived from the original IntoValue dataset [35]. Dissertations were excluded from publication-based metrics.

**Open Access (OA) status.** To determine the OA status of trial results publications, we queried the Unpaywall database via its API on 1 November 2022 using UnpaywallR and assigned one of the following statuses: gold (openly available in an OA journal), hybrid (openly available in a subscription-based journal), green (openly available in a repository), bronze (openly available on the journal page but without a clear open license), or closed. As publications can have several OA versions, we applied a hierarchy such that only one OA status was assigned to each publication, in descending order: gold, hybrid, green, bronze, and closed.

## Interactive dashboard

We developed an interactive dashboard to present the outcome of these assessments at the institutional level in an accessible way to the UMC leadership and the wider research community. The dashboard was developed with the Shiny R package (version 1.6.0) [38] based on an initial version developed by NR for the Charité –Universitätsmedizin Berlin [15]. The dashboard was shaped by interviews with UMC leadership, support staff, funders, and experts in

**Table 1. Sample considered and measured outcome for the trial transparency practices in the dashboard.**

| Section | Practice | Sample (denominator) | What was measured |
|---|---|---|---|
| Trial registration | Prospective registration | **DRKS (IntoValue)**: trials in the IntoValue cohort and registered in DRKS (interventional, led by a German UMC, completion date between 2009–2017, and considered as complete based on study status in the registration) with a start date in the registry | Was the trial registered in the same month or in a previous month to the trial start date? |
| | | **ClinicalTrials.gov**: recent cohort of trials registered in ClinicalTrials.gov (interventional, led by a German UMC, start date between 2006–2018, and considered as complete based on study status in the registration) | |
| | Reporting of the TRN in results publications | All trials in the IntoValue cohort (registered in ClinicalTrials.gov or DRKS) with a manuscript publication and a PubMed Identifier (detection of TRN in abstract) or for which the full text could be retrieved (detection of TRN in full text) | For trials with a manuscript publication, was the TRN reported (a) in the abstract; (b) in the full text? |
| | Publication link in the trial registry | All trials in the IntoValue cohort (registered in ClinicalTrials.gov or DRKS) with a manuscript publication and a DOI or a PubMed Identifier | For trials with a manuscript publication, is said publication linked in the registration? |
| Trial reporting | Summary results reporting in the EUCTR | Due trials listed on the EU Trials Tracker (and therefore registered in the EUCTR) with a sponsor name corresponding to one of the included UMCs | How many due trials registered in the EUCTR have reported summary results in the registry? |
| | Summary results reporting in ClinicalTrials.gov or DRKS | All trials in the IntoValue cohort (registered in ClinicalTrials.gov or DRKS) | How many due trials registered in ClinicalTrials.gov or DRKS have reported summary results in the registry? |
| | Results reporting within 2 and 5 years of trial completion (summary results or manuscript publication) | All trials in the IntoValue cohort (registered in ClinicalTrials.gov or DRKS). **Reporting as summary results:** only trials with a follow-up time of 2 and 5 years from trial completion to the registry download date were included. **Reporting as a manuscript publication:** only trials with a follow-up time of 2 and 5 years from trial completion to the manual publication search date were included. **Reporting as summary results or manuscript publication:** only trials with a follow-up time of 2 and 5 years from (1) trial completion to the registry download date AND (2) trial completion to the manual publication search date were included. | How many trials have reported results within 2 and 5 years of trial completion? The following reporting routes were considered: summary results, manuscript publication, summary results or manuscript publication |
| OA | OA status | Unique results publications from the IntoValue cohort of trials (registered in ClinicalTrials.gov or DRKS) with a DOI and a publication date in Unpaywall | Of all trial publications, how many are openly accessible and via which route (gold OA, hybrid OA, green OA, or bronze OA)? |

Overview of the transparency practices assessed and included in the dashboard, along with the sample considered, and the measured outcome. See **S5 and S8 Supplements** for more detailed information.

DOI, Digital Object Identifier; DRKS, German Clinical Trials Register; EUCTR, EU Clinical Trials Register; OA, Open Access; TRN, Trial Registration Number; UMC, University Medical Center.

responsible research who provided feedback on a prototype version [39]. This feedback led to the inclusion of several features to facilitate the interpretation of the data and contextualize the assessed transparency practices. The code underlying the dashboard developed in this study is openly available in GitHub under an AGPL license (https://github.com/quest-bih/clinical-dashboard) and may be adapted for further use.

## Analysis

We generated descriptive statistics on the characteristics of the trials and the transparency practices, all of which are displayed in the dashboard. We report proportions across UMCs (e.g., "Start" page) and per UMC broken down by start year (prospective registration only),

completion year, publication year (open access), and registry (publication link in the registry, summary results reporting). We did not test specific hypotheses.

## Software, code, and data

Data processing was performed in R (version 4.0.5) [40] and Python 3.9 (Python Software Foundation, Wilmington, Delaware, USA). With the exception of summary results reporting in the EUCTR (data available via the EU Trials Tracker), all the data processing steps involved in generating the dataset displayed in this dashboard are openly available in GitHub: https://github.com/maia-sh/intovalue-data/releases/tag/v1.1. The data displayed in the dashboard are available in OSF [41] and in the dashboard Datasets page. Raw data obtained from trial registries are openly available in Zenodo [42]. This study is reported as per the Strengthening the Reporting of Observational Studies in Epidemiology (STROBE) guideline for cross-sectional studies (**S6 Supplement**).

## Results

### Characteristics of trials

The IntoValue dataset that this study is based on includes interventional trials registered in ClinicalTrials.gov or DRKS, led by a German UMC, and reported as complete between 2009 and 2017 (*n* = 3,113). Trials were found for 35 out of 38 UMCs searched. After downloading updated registry data for trials in this cohort, we excluded 91 trials based on our exclusion criteria (study completion date before 2009 or after 2017, *n* = 73; not considered as complete per study status, *n* = 16; not interventional, *n* = 2). After removal of duplicates, this led to 2,895 trials that served as the basis for most metrics (**Fig 2**). For prospective registration in ClinicalTrials.gov, we used a more recent cohort of interventional trials registered in ClinicalTrials.gov, led by a German UMC, started between 2006 and 2018, and considered as

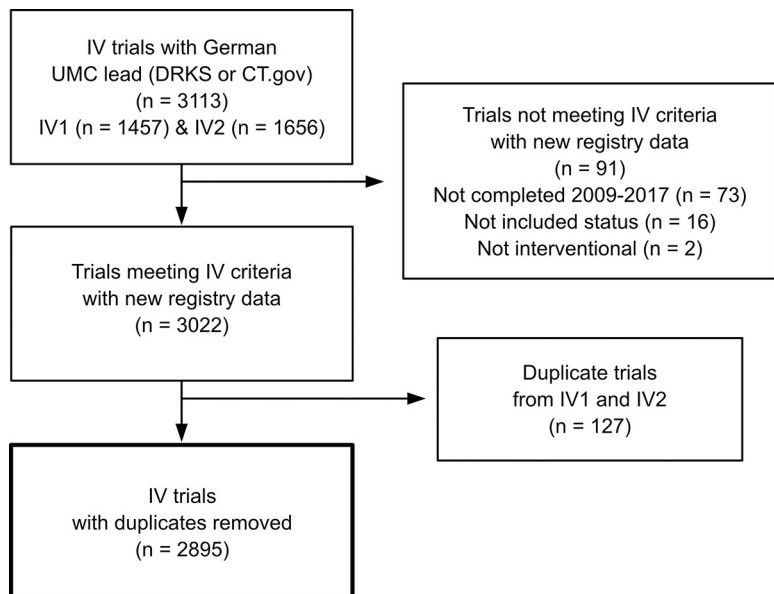

**Fig 2. Trial screening.** Flowchart of the trial screening (IntoValue). The box with the thicker contour highlights the starting point of the trial screening for other registry-based metrics (see Flowcharts 1–3 in S8 Supplement). CT.gov, ClinicalTrials.gov; DRKS, German Clinical Trials Register; IV, IntoValue; UMC, University Medical Center.

complete per study status in the registry ($n$ = 4,058). After applying our inclusion criteria, this sample included 3,618 trials. **S7 Supplement** provides an overview of the characteristics of included trials stratified by registry. **S8 Supplement** provides flow diagrams of the trial and publication screening for each metric.

## Evaluation of trial registration and reporting practices

We developed an interactive dashboard (https://quest-cttd.bihealth.org/) to display the results of the evaluation of trial registration and reporting across UMCs. In the following, we highlight some of these results. More extensive evaluations of some of these practices are reported in separate publications, such as results reporting of trials [9,11] and links between trial registration and results publications [10].

## Trial registration

**Prospective registration**: The proportion of trials led by German UMCs that were prospectively registered increased in both ClinicalTrials.gov and DRKS over the period considered. Of 178 trials registered in ClinicalTrials.gov and started in 2006, 58 (33%, 95% confidence interval 26% to 40%) were registered prospectively. A little more than a decade later, 144 of 193 (75%, 95% confidence interval 68% to 80%) trials started in 2018 were registered prospectively. Trials registered in DRKS followed a similar trend: While none of the 44 (0%, 95% confidence interval 0% to 10%) trials started between 2006 and 2008 were prospectively registered, this increased to 19 of 24 (79%, 95% confidence interval 57% to 92%) for trials started in 2017 (**S9 Supplement**). Among clinical trials registered in ClinicalTrials.gov, the median per-UMC rate of prospective registration ranged from 30% ($n$ = 17/56) to 68% ($n$ = 127/186) with a median of 55% and a standard deviation of 8%. Per-UMC rates of prospective registration in DRKS ranged from 0% ($n$ = 0/1) to 75% ($n$ = 15/20) with a median of 44% and a standard deviation of 15%.

**Reporting of a TRN in publications.** Of the 1,895 registered trials with a publication indexed in PubMed, 714 (38%, 95% confidence interval 35% to 40%) reported a TRN in the publication abstract. In turn, 1,136 of 1,893 registered trials for which the full text was available reported a TRN in the publication full text (60%, 95% confidence interval 58% to 62%) (**S9 Supplement**). Only 476 of 1,893 (25%, 95% confidence interval 23% to 27%) of trials reported a TRN in both the abstract and full text of the publication as per the ICMJE and Consolidated Standards of Reporting Trials (CONSORT) guidelines. The per-UMC rate at which clinical trial publications reported a TRN in the abstract ranged from 17% ($n$ = 13/75) to 56% ($n$ = 23/41) with a median of 38% and a standard deviation of 8%. The per-UMC rate at which clinical trial publications reported a TRN in the full text was higher, ranging from 43% ($n$ = 41/95) to 76% ($n$ = 32/42) with a median of 61% and a standard deviation of 7%.

**Publication links in the registry.** Of 1,493 trials registered in ClinicalTrials.gov with a publication, 861 (58%, 95% confidence interval 55% to 60%) had a link to the publication in the registration. In turn, only 111 of 474 trials registered in DRKS with a publication (23%, 95% confidence interval 20% to 28%) had a link to the publication in the registration. Among trials registered in ClinicalTrials.gov with a publication, the per-UMC rate of publication links in the registration ranged from 32% ($n$ = 12/37) to 88% ($n$ = 28/32) with a median of 56% and a standard deviation of 12%. Among trials registered in DRKS with a publication, the per-UMC rate of publication links in the registration ranged from 0% ($n$ = 0/7) to 45% ($n$ = 5/11) with a median of 23% and a standard deviation of 13%.

## Trial reporting

**Summary results reporting.** We first assessed how many of the trials registered in ClinicalTrials.gov or DRKS had summary results in the registry. The cumulative proportion of trials that reported summary results has stagnated at low levels between 2009 and 2017. Only 191 of all 2,253 (8%, 95% confidence interval 7% to 10%) trials registered in ClinicalTrials.gov, and 20 of all 642 (3%, 95% confidence interval 2% to 5%) trials registered in DRKS had summary results in the registry (**S9 Supplement**). Per-UMC summary results reporting rates for all trials ranged between 0% (*n* = 0/42) and 32% (*n* = 8/25) (median of 7% and a standard deviation of 7%) for ClinicalTrials.gov, and between 0% (*n* = 0/23) and 50% (*n* = 7/14) (median of 0% and a standard deviation of 9%) for DRKS. In contrast, reporting of summary results in the EUCTR was higher and increased over time: In almost 2 years, results reporting for due trials almost doubled from 41% (*n* = 223/541, 95% confidence interval 37% to 46%) in December 2020 to 79% (*n* = 647/813, 95% confidence interval 77% to 82%) in October 2022 (EU Trials Tracker) (**S9 Supplement**). At the time of data collection (November 2022), per-UMC summary results reporting rates in the EUCTR ranged between 0% (*n* = 0/1) and 100% (*n* = 14/14) across all included UMCs with a median of 82% and a standard deviation of 30%.

**Timely reporting of results (2- and 5-year reporting rates).** Next, we assessed how many trials registered in ClinicalTrials.gov or DRKS reported results in a timely manner. Reporting guidelines and German research funders have called on clinical trials to report (a) summary results in the registry within 12 and 24 months of trial completion and (b) results in a manuscript publication within 24 months of trial completion [2,43–45]. We therefore considered 2 years as timely reporting for both reporting routes. Of 2,892 trials registered in ClinicalTrials.gov or DRKS with a 2-year follow-up period for reporting results as either summary results or a manuscript publication, 1,198 (41%, 95% confidence interval 40% to 43%) had done so within 2 years of trial completion.

While the 5-year reporting rate was unsurprisingly higher, 505 of 1,619 trials (31%, 95% confidence interval 29% to 34%) registered in ClinicalTrials.gov or DRKS with 5-year follow-up between trial completion and the manual publication search had not reported results as a journal publication within 5 years of trial completion. Publication in a journal was the dominant route of reporting results, with summary results reporting rates below 10% across all completion years and follow-up periods. Per-UMC reporting rates as a manuscript publication ranged between 15% (*n* = 7/46) and 58% (*n* = 19/33) (2-year rate, median 39%, standard deviation 9%) and between 50% (*n* = 24/48) and 87% (*n* = 13/15) (5-year rate, median 70%, standard deviation 8%). Per-UMC reporting rates as summary results ranged between 0% (*n* = 0/76) and 14% (*n* = 6/43) (2-year rate, median 4%, standard deviation 4%) and between 0% (*n* = 0/72) and 21% (9/42) (5-year rate, median 5%, standard deviation 5%).

**OA status.** The proportion of trial results publications that were openly accessible (gold, hybrid, green, or bronze) increased from 42% in 2010 (*n* = 16/38, 95% confidence interval 27% to 59%) to 74% in 2020 (*n* = 72/97, 95% confidence interval 64% to 82%) (**S9 Supplement**). Across all publication years, 891 of 1,920 (46%, 95% confidence interval 44% to 49%) trial publications were neither openly accessible via a journal nor an OA repository based on Unpaywall. Per-UMC rates of trial results publications that were OA ranged from 26% (*n* = 10/38) to 72% (*n* = 23/32) with a median of 55% and a standard deviation of 10%.

### Interactive dashboard

The key outcome of this paper is an interactive and openly accessible dashboard to visualize adherence to the aforementioned best practices for trial registration and reporting across German UMCs: https://quest-cttd.bihealth.org/. The dashboard displays the data in 3 ways: (a) assessment across all UMCs (national dashboard; see a screenshot in **Fig 3**); (b) comparative assessment between UMCs; and (c) UMC-specific assessment (see a screenshot for one UMC in **S10 Supplement**).

To allow for a better interpretation of the data displayed in the dashboard, absolute numbers are displayed in all plots as mouse-overs. A description of the methods and limitations of each metric is also provided next to each plot, with more detailed information in the Methods page. A FAQ page addresses general considerations raised in interviews with relevant stakeholders [39]. These interviews highlighted the importance of an overall narrative justifying the choice of metrics included. We therefore designed an infographic of relevant laws and guidelines to contextualize the clinical transparency metrics included in the dashboard (adapted from **Fig 1**).

## Discussion

Concerns about delayed and incomplete results reporting in clinical research and other sources of research waste have triggered debate on incentivizing individual researchers and UMCs to adopt more responsible research practices [20,22,23]. Here, we introduced the methods and results underlying a dashboard for clinical trial transparency, which provides actionable information on UMCs' performance in relation to established registration and reporting practices and thereby empowers their efforts to support improvement. This dashboard approach for clinical trial transparency at the level of individual UMCs serves to (a) inform institutions about their performance and set this in relation to national and international transparency guidelines and funder mandates, (b) highlight where there is room for improvement, (c) trigger discussions across relevant stakeholder groups on responsible research practices and their role in assessing research performance, (d) point to success stories and facilitate knowledge sharing between UMCs, and (e) inform the development and evaluation of interventions that aim to increase trial transparency.

### Trends in trial transparency

The dashboard displays progress over time and allows the data to be explored in different ways. While the upward trend for several practices (e.g., prospective registration, OA) is encouraging, there is much room for improvement with respect to established guidelines for clinical trial transparency. For example, less than half (45%) of trials registered in ClinicalTrials.gov or DRKS and completed in 2017 reported results in a manuscript publication within 2 years of trial completion as per WHO and funder recommendations [2,43,44]. We observed a striking difference in the cumulative proportion of summary results reporting of drug trials registered in the EUCTR compared with trials registered in ClinicalTrials.gov and DRKS. The uptake of summary results reporting in the EUCTR likely reflects the combined impact of the EU legal requirement for drug trials to report summary results within 12 months [45], the launch of the EU Trials Tracker and subsequent academic initiatives to increase reporting rates [8,18], as well as media attention [46]. This suggests that audits of compliance with respect to established guidelines and further awareness raising may also have the potential to increase results reporting rates of other types of trials.

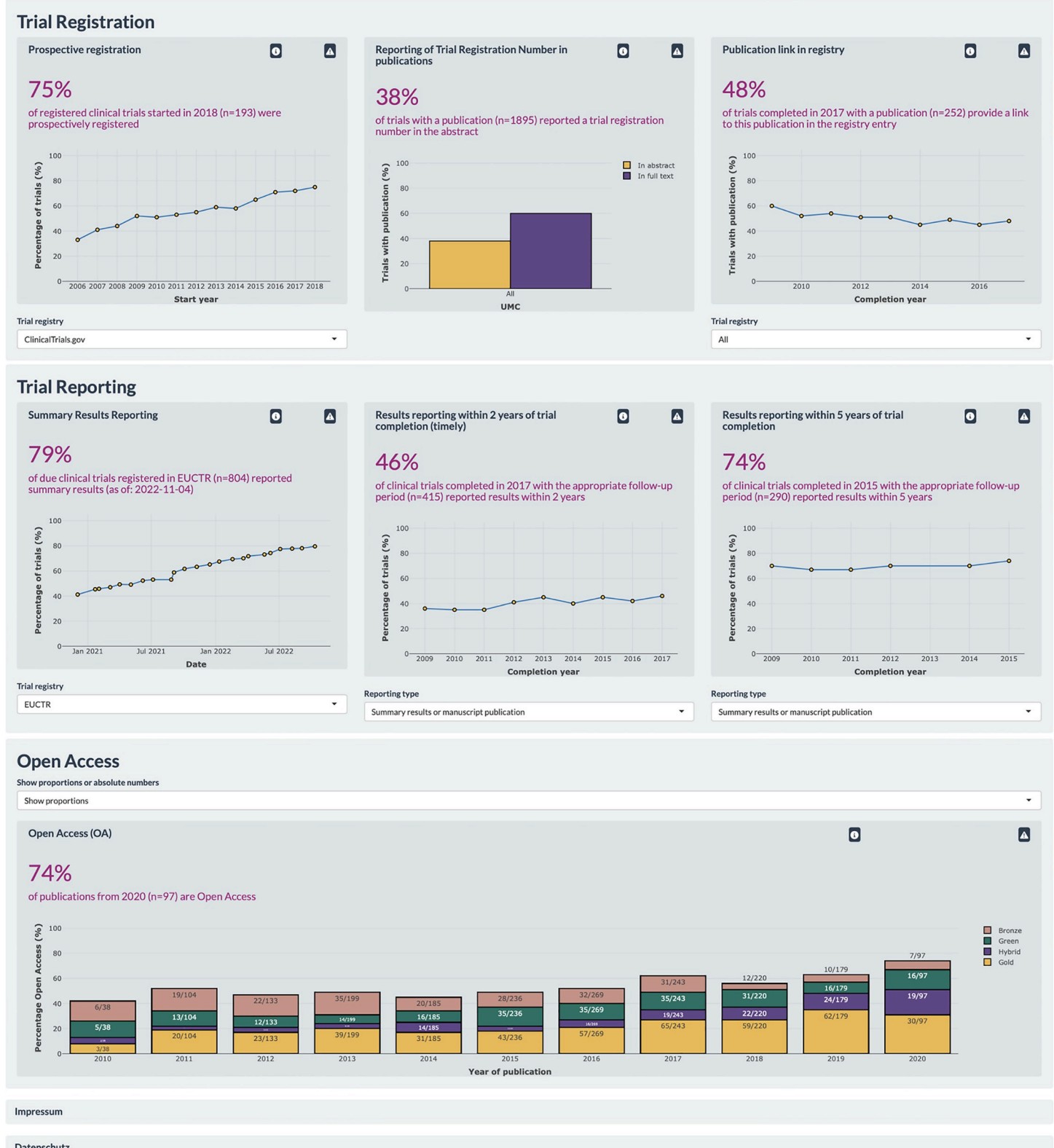

**Fig 3. Screenshot of the home ("Start") page of the dashboard for clinical trial transparency.** Assessment of 7 registration and reporting practices across all included German UMCs (8 November 2022).

### Actionable areas for stakeholders

Some of the practices included in this dashboard can still be addressed retroactively, such as linking publications in the trial registration (realized for 49% of trials with a publication). These constitute actionable areas for improvement that UMCs can contribute to by providing education, support, and incentives. One important way to incentivize UMCs in this regard is to make responsible research practices part of internal and external quality assessment procedures. Other stakeholders such as funders, journals and publishers, registries, and bibliographic databases should complement these activities by reviewing compliance with their policies as well as applicable guidelines and/or laws. Salholz-Hillel and colleagues, for example, outlined specific recommendations for each stakeholder to improve links between trial registrations and publications [10]. UMCs and their core facilities for clinical research can, for example, use the data linked to the dashboard to inform principal investigators about the transparency of their specific trials. We are currently finalizing such a "report card" approach at the Charité - Universitätsmedizin Berlin [47].

### Scalability beyond German UMCs

The datasets and methods used in this study can be scaled: This has been demonstrated in another European country (Poland) [48] and is currently underway in California, USA [49]. While the generation of the underlying dataset of clinical trials and associated results publications involves manual checks (approximately 10 person-hours per 100 trials), the assessment of transparency practices is largely automated. Institutions in possession of an in-house cohort of clinical trial registry numbers and persistent identifiers (e.g., Digital Object Identifier (DOI)) from matched journal publications, however, could achieve results more quickly. The code to create the dashboard is openly available and can be adapted to other cohorts.

### Stakeholder and community engagement

The uptake of this dashboard approach by UMCs and other stakeholders depends on their respective attitudes and readiness. We previously solicited stakeholders' views on an institutional dashboard with metrics for responsible research. While interviewees considered the dashboard helpful to see where an institution stands and to initiate change, some pointed to the challenge that making such a dashboard public might risk incorrect interpretation of the metrics and harm UMCs' reputation [39]. While similar challenges with interpretation and reputation apply to current metrics for research assessment (e.g., impact factors and third-party funding), this stakeholder feedback demonstrates the need for community engagement when introducing novel strategies for research assessment. In this regard, a Delphi study was performed to reach consensus on a core outcome set of open science practices within biomedicine to support audits at the institutional level [50]. A detailed comparative assessment of existing monitoring initiatives and lessons learned could further support these efforts.

### Updates and further development of the dashboard

We are planning regular updates of the registry data for trials already in the dashboard, as well as the inclusion of more recent cohorts of trials with at least 2 years follow-up (e.g., trials completed 2018 to 2021 assessed in 2023). Besides these updates, further transparency practices may be integrated into the dashboard in the future, e.g., dissemination of results as preprints, the use of self-archiving to broaden access to results [51], adherence to reporting guidelines [3], or data sharing [52]. Beyond transparency, other potential metrics could reflect the number of discontinued trials [53] or the proportion of trials that inform clinical practice [54]. The

development of such metrics should acknowledge the availability of standards and infrastructure pertaining to the underlying practices [23] and differences between study types and disciplines [27]. Future versions of the dashboard may also display additional subpopulation comparisons, such as different clinical trial registries or UMC particularities [55].

## Limitations

A limitation of this study is that inaccurate or outdated registry data (e.g., incorrect completion dates or trial status) may have impacted the assessment of transparency practices described in this study. To mitigate this limitation, we updated the registry data with the most recent data we could obtain. The update-related changes suggest no systematic bias in the comparison across UMCs. Another limitation is that the trial dataset may contain more cross-registrations than we identified. For the aforementioned "report card" project, we manually verified 168 trials and found only 2 missed cross-registrations (1%). We therefore believe that missed cross-registrations represent only a small portion of our sample. Moreover, the assessment of each practice in the dashboard applies to a specific subset of trials or publications and comes with unique limitations, largely resulting from challenges associated with manual or automated methods (outlined in more detail in **S5 Supplement**). More generally, the dashboard focuses on interventional trials registered in ClinicalTrials.gov or DRKS and does not display how German UMC drug trials only registered in the EUCTR perform on established transparency practices (except for summary results reporting in the registry). We are considering including all drug trials in the EUCTR conducted by German UMCs in future developments of the dashboard.

## Conclusions

UMCs play an important role in fostering clinical trial transparency but face challenges doing so in the absence of baseline assessments of current practice. We assessed adherence to established practices for clinical trial registration and reporting at German UMCs and communicated the results in the form of an interactive dashboard. We observed room for improvement across all assessed practices, some of which can still be addressed retroactively. The dashboard provides actionable information to drive improvement, facilitates knowledge sharing between UMCs, and informs the development of interventions to increase research transparency.

## Supporting information

**S1 Supplement. Use of automated vs. manual approaches across methods.**
(PDF)

**S2 Supplement. Inclusion and exclusion criteria.**
(PDF)

**S3 Supplement. Selected sponsor names in the EU Trials Tracker.**
(PDF)

**S4 Supplement. Sources for Fig 1.**
(PDF)

**S5 Supplement. Detailed methods and limitations of registration and reporting metrics.**
(PDF)

**S6 Supplement. STROBE checklist for cross-sectional studies.**
(PDF)

**S7 Supplement. Characteristics of included trials.**
(PDF)

**S8 Supplement. Flow diagrams of the trial and publication screening.**
(PDF)

**S9 Supplement. Screenshots of the "Start" page of the dashboard.**
(PDF)

**S10 Supplement. Screenshot of the "One UMC" page of the dashboard.**
(PDF)

## Acknowledgments

We would like to acknowledge Tamarinde Haven and Martin Holst for their valuable input that shaped the dashboard. We acknowledge financial support from the Open Access Publication Fund of Charité – Universitätsmedizin Berlin and the German Research Foundation (DFG).

## Author Contributions

**Conceptualization:** Delwen L. Franzen, Benjamin Gregory Carlisle, Maia Salholz-Hillel, Daniel Strech.

**Data curation:** Delwen L. Franzen, Benjamin Gregory Carlisle, Maia Salholz-Hillel.

**Funding acquisition:** Daniel Strech.

**Investigation:** Delwen L. Franzen, Benjamin Gregory Carlisle, Maia Salholz-Hillel.

**Methodology:** Delwen L. Franzen, Benjamin Gregory Carlisle, Maia Salholz-Hillel, Daniel Strech.

**Project administration:** Delwen L. Franzen, Daniel Strech.

**Software:** Delwen L. Franzen, Benjamin Gregory Carlisle, Maia Salholz-Hillel, Nico Riedel.

**Supervision:** Daniel Strech.

**Visualization:** Delwen L. Franzen, Benjamin Gregory Carlisle, Maia Salholz-Hillel, Nico Riedel.

**Writing – original draft:** Delwen L. Franzen, Daniel Strech.

**Writing – review & editing:** Delwen L. Franzen, Benjamin Gregory Carlisle, Maia Salholz-Hillel, Nico Riedel, Daniel Strech.

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
