## [Editor Report · Decision Letter 0]

3 May 2022

Dear Dr Franzen, 

Thank you for submitting your manuscript entitled "An institutional dashboard to drive clinical research transparency: An open and scalable case study at University Medical Centers" for consideration by PLOS Medicine.

Your manuscript has now been evaluated by the PLOS Medicine editorial staff and I am writing to let you know that we would like to send your submission out for external peer review.

Please re-submit your manuscript within two working days, i.e. by May 05 2022 11:59PM.

Kind regards,

Caitlin Moyer, Ph.D.

Associate Editor

PLOS Medicine

---

## [Decision Letter · Decision Letter 1]

13 Sep 2022

Dear Dr. Franzen,

Thank you very much for submitting your manuscript "An institutional dashboard to drive clinical research transparency: An open and scalable case study at University Medical Centers" (PMEDICINE-D-22-01430R1) for consideration at PLOS Medicine. 

Your paper was evaluated by a senior editor and discussed among all the editors here. It was also discussed with an academic editor with relevant expertise, and sent to four independent reviewers, including a statistical reviewer. The reviews are appended at the bottom of this email and any accompanying reviewer attachments can be seen via the link below:

[LINK]

In light of these reviews, I am afraid that we will not be able to accept the manuscript for publication in the journal in its current form, but we would like to consider a revised version that addresses the reviewers' and editors' comments. Obviously we cannot make any decision about publication until we have seen the revised manuscript and your response, and we plan to seek re-review by one or more of the reviewers. 

We expect to receive your revised manuscript by Oct 04 2022 11:59PM. Please email us (plosmedicine@plos.org) if you have any questions or concerns.

We look forward to receiving your revised manuscript. 

Sincerely,

Caitlin Moyer, Ph.D.

Associate Editor

PLOS Medicine

plosmedicine.org

From the Academic Editor:

1. The authors should pay attention to avoid citation shortcuts. In some occasions, it makes the manuscript difficult to follow and blurs the line between what is new and what was already published. This being said, the paper is very well written, all previous papers are appropriately cited, a web appendix, and online materials are available.

2. Authors also cite various initiatives (trial tracker, the French Monitor for Open Science, etc). I would be very interested in a detailed comparative assessment of these existing initiatives (in terms of metrics used, timeframe, etc.). Authors cite an initiative about developing a core outcome set. Describing all outcomes used by existing initiatives may inform the development of such a core outcome set.

3. The dashboard presented in Figure 2 displays mean values. Of course, this is important. But as this is a dashboard that aims to explore various institutions, one would be interested in values of all institutions (including the max and the min). I wonder if it would be possible to display this information.

4. One major problem of these monitors is that they are based on data which are likely biased. I'm not sure that trials registries are appropriately updated by all UMCs, e.g. regarding study status. It may introduce a lot of bias in the output, and this bias could be differential when one compares UMCs. This possible issue should be discussed in more detail as it is a major limitation of this comparative approach.

5. Authors may want to discuss their results in line with those of Speich et al. PLOS Medicine. 2022 and Thiele et al. Plos One 2022 ;

6. Introduction:

– I would stress that there is a very favorable environment for these initiatives

–More emphasis on DORA and the Hong Kong principles is needed ;

– I would also welcome more emphasis on the European approach for rewarding reproducible research practices and Open Science ;

–I would also cite the UNESCO approach of Open Science ;

7. Methods:

– I would provide more details on the automatic versus manual part of the extraction as this probably a very big problem for reproducibility regarding data extraction ;

–A citation is needed concerning the list of German UMC that was used ;

.--p6. 115, a citation is needed after has been described previously ;

–Again, the interactive dashboard is the most important and original part and it needs more emphasis ;

8. Results:

–The various flow charts in the supplements are very informative but difficult to follow. I was wondering if a single flow chart with various branches would be a better figure (I acknowledge that this part is very challenging to represent adequately) ;

9. Discussion:

– Author should elaborate a little bit more on the frequency of updates etc.

–Authors should also discuss the implication of this dashboard, for instance, should the trial registry implement such a dashboard directly? Which international organisation may be interested in using at a large scale such a dashboard? How difficult would it be to transfer such a dashboard from Germany to the rest of the world?

–Authors should stress that implementation of such dashboard will not guarantee that UMCs will be more transparent. What kind of research could be envisioned to answer this question? I would love to see more elaboration on this point.

Other editorial points:

10. Title: Please revise your title according to PLOS Medicine's style. Your title must be nondeclarative and not a question. It should begin with main concept if possible. "Effect of" should be used only if causality can be inferred, i.e., for an RCT. Please place the study design ("A randomized controlled trial," "A retrospective study," "A modelling study," etc.) in the subtitle (ie, after a colon).

11. Data availability statement: Please also include a link to the dashboard, where it will be made publicly available.

12. Abstract: Please combine the Methods and Findings sections into one section, “Methods and findings”.

13. Abstract: Line 11: Please define DRKS at first use.

14. Abstract: Methods and Findings: Please describe how adherence to established transparency practices was assessed.

15. Abstract: Methods and Findings: Please ensure that all numbers presented in the abstract are present and identical to numbers presented in the main manuscript text.

16. Abstract: Methods and Findings: In the last sentence of the Abstract Methods and Findings section, please describe the main limitation(s) of the study's methodology.

17. Abstract: Conclusions: * Please address the study implications without overreaching what can be concluded from the data; the phrase "In this study, we observed ..." may be useful.

18. Author summary: At this stage, we ask that you include a short, non-technical Author Summary of your research to make findings accessible to a wide audience that includes both scientists and non-scientists. The Author Summary should immediately follow the Abstract in your revised manuscript. This text is subject to editorial change and should be distinct from the scientific abstract. Please see our author guidelines for more information: https://journals.plos.org/plosmedicine/s/revising-your-manuscript#loc-author-summary

19. Main text: Please place in-text citations for references within square brackets. Where multiple references are indicated, please do not include spaces within brackets.

20. Main text: Please define all acronyms at first use in the text.

21. Methods: For all observational studies, in the manuscript text, please indicate: (1) the specific hypotheses you intended to test, (2) the analytical methods by which you planned to test them, (3) the analyses you actually performed, and (4) when reported analyses differ from those that were planned, transparent explanations for differences that affect the reliability of the study's results. If a reported analysis was performed based on an interesting but unanticipated pattern in the data, please be clear that the analysis was data-driven.

22. Methods: Did your study have a prospective protocol or analysis plan? Please state this (either way) early in the Methods section.

23. Methods: Please report your study according to the relevant guideline, which can be found here: http://www.equator-network.org/

Please ensure that the study is reported according to the STROBE or other guideline, and include the completed STROBE (or other) checklist as Supporting Information. When completing the checklist, please use section and paragraph numbers, rather than page numbers. Please add the following statement, or similar, to the Methods: "This study is reported as per the Strengthening the Reporting of Observational Studies in Epidemiology (STROBE) guideline (S1 Checklist)." (Or please adapt to the most appropriate guideline for your study.)

24. Methods Line 93-95: Although described elsewhere, it would be helpful to have the methods for sampling and tracking results described here.

25. Results: Please present numerators and denominators for percentages.

26. Discussion: Please present and organize the Discussion as follows: a short, clear summary of the article's findings; what the study adds to existing research and where and why the results may differ from previous research; strengths and limitations of the study; implications and next steps for research, clinical practice, and/or public policy; one-paragraph conclusion.

27. Lines 336-349: Please remove the Funding, Author contributions, and Competing Interests sections from the main text. Please be sure all information is completely and accurately entered into the manuscript submission system.

28. References: Please use the "Vancouver" style for reference formatting, and see our website for other reference guidelines https://journals.plos.org/plosmedicine/s/submission-guidelines#loc-references

29. Table 1: In the legend, please define all abbreviations used in the table.

30. Supplement S2: In the legend, please define all abbreviations used in the table.

31. Supplement S4: In the legend, please define all abbreviations used in the table.

32. Supplement S5: Please define all abbreviations used in the flowcharts.

Comments from the reviewers:

Reviewer #1: 1. Please report p-values and 95% confidence interval in the abstract, and results section

2. Line 162/165: I suggesting adding more details about why 3790=2915=875 studies were excluded

3. The analysis mainly focused on the description of the overall "clinical transparency" over time. What are the underlying reasons for the changes overtime? Is it possible to examine the heterogeneity in the outcomes in study characteristics(e.g. individual UMC, size of study sample, disease category, results of the trial [ effective or not])

Reviewer #2: I would like to congratulate the authors on a thorough review of trial registration and results posting in the German Academic research environment. Although we could not see it completely, I also applaud the creation of a Dashboard for future data display.

I have a few comments for the authors' consideration.

The authors need to carefully distinguish disclosure (which is what they discuss) from transparency. Disclosure is a necessary but not sufficient step in transparency and open science. As the authors point out the disclosures need to findable and accessible to meet the FAIR principles. It would be good for the authors to reference FAIR upfront as they underpin open science in general. You have clearly outlined the principles on page 3 line 38.

There needs to be care in comparing results to clinical trials.gov. Note that the requirement to register and post results does not apply to Phase I trials and that might or might not affect the denominator. Also, before 2017 the results requirements only applied to trials of approved medicines so again the denominator for posted results may not be ALL trials. It is true that the ICMJE criteria were broader than that and while of great influence were not strictly speaking legal requirements. 

P 3 line 40 You refer to "registered" with reference to results- I assume you mean there are results posted for trials that were not registered?

P 5 - It would be useful to define what you mean by "prospective". For CT.Gov the requirement is to register no later than 21 days after the first patient is enrolled. Is that the definition you are using?

P 8 line 163. Under results when you use the term "registered" are you using the "prospective" definition or all? In other places you are specific.

P 9 line 192: I agree part of findable means referring to the TRN. It is disappointing journals are not enforcing that and perhaps that needs to called out.

In the discussion you mention for the first time in the paper preclinical research and while the statement may be true you have not addressed that anywhere else and I suggest you might delete that here as it is a new concept. You also refer to "clinical research" in reference to your dashboard, but it is really clinical trials, correct? There are many discussions surrounding the registration of observation studies and I assume your dashboard is restricted to interventional trials as noted in the beginning?

Reviewer #3: PMedicine D-22-01430

1. The authors describe the development of a dashboard to illuminate execution of commitments to transparency in clinical trials. Note that many of the substantive results have been published previously, including trial registration, results reporting of trials, linkage of publications to trial registration and others. Therefore, the description and utility of the dashboard is the substantive new contribution, but it is rather briefly described (e.g., one paragraph in the manuscript). The description, development, user engagement, potential adaptations etc should be more fully described.

2. It would be very important to deduplicate the trials that are reported both in Clinicaltrials.gov and DRKS and/or divide the results by those that are registered and reported in one but not the other or both. The expectations and nature of the trials may differ.

3. Did the authors analyze by type of trial (regulated, drug/biologic/device/behavioral, industry v other, etc.) differences in registration, results reporting, linkage of registration to publications, and the other outcome measures? If not, why not?

4. The authors have used a dataset, published previously, that includes trials from 2009-2017. The standards and cultural expectations of transparency, including registration and results reporting, ORCID ID associations, publication preprint and access, have changed significantly in the last 5 years. It is unclear, if this approach is to be scalable and replicable as claimed, why the data was not updated to current state or at a minimum, a more recent end date (July 2021). Only the metric for publications would need to be adjusted for time, given the 2 year window for publication. All other metrics could be shown. A 5 year delay (or, as apparent annual updates are planned that include data from 3 years prior) is hardly relevant to current activities either by institution or sponsor.

a. It is unclear why results reporting at 5 years is less than at 2 years. If a trial is reported within 2 years, isn't it also included in the 5 year number? It is certainly reported at 5 years if reported at 2.

I suggest the authors update the data files and statistics, and replace the results as discussed below.

5. Similarly, for prospective registration of clinical trials, the authors used "a more recent cohort of interventional trials" 2006-2018. The field has changed dramatically since then, so why not use a truly recent cohort? E.g. ending 2021? Pooled statistics over that much time as described in the manuscript is not helpful.

6. A major deficiency of the manuscript is the failure to describe the trends in the reported outcomes (trial registration, results reporting, linkage of registration to publications, etc.) as the substance of the results section. There have been significant changes since 2006; pooling all the results necessarily decreases the utility of the findings. Coupled with the lack of recent data, this is highly problematic. The data should be reported by year, and by institution, and interpreted at more granular level of analysis. Graphs are shown in the figures, but the trends are not the focus of the results described.

7. The authors describe variability between and among UMC for trial registration but do not investigate that further. Are the trials that are not 'pre' registered applicable clinical trials? Are the trials not registered on Clinicaltrials.gov nevertheless registered on DRKS? What are the implications of interventional trials that are registered, but not pre-registered—did they need to be preregistered? Is the coding correct?

8. The authors claim that this manuscript describes "stakeholder engagement to provide UMCs with actionable information on trial transparency in an accessible and interactive format, and thereby empower their efforts to support improvement." This is not described in the current manuscript. Stakeholder engagement and how the stakeholders used or did not use the data to understand or change current practices are not described.

9. The authors should appreciate that some of their suggestions are not accomplished with "minimal effort" as claimed. A simple update (retrospectively linking publications in the trial registration) in Clinicaltrials.gov "opens" the trial record, and the entire record must then be updated to current requirements. The number of changes to requirements and the effort to do so are significant and a disincentive to retrospective changes. 

10. Note that https://quest-cttd.bihealth.org requires a log in, which, given confidentiality provisions for reviewers, is not possible. The manuscript does alert to the fact that it will be open access upon publication, but it makes it difficult to review the utility, user-experience, and impact in the absence of access. Authors' claims to its novelty and importance (e.g., "provid[ing[actionable information") cannot be evaluated.

Reviewer #4: 

This is a useful and interesting article on a novel Dashboard that provides seven parameters on the quality of trials registration and reporting for German University Medical Centres (UMCs). The centrepiece of the article is really the Figure to showing the Dashboard for all UMC's. However, the text also provides some interesting and novel data on registration and reporting features, such as the proportion of published trials that report the trial registration number in both abstract and text, and the number of published trials which have provided a summary on the trials registry - which is appallingly low. While the article made interesting reading, I was left wondering what those in the University medical centres might think of this dashboard? Some suggestions for improvement:

1/ It would help readers if the numbers in the text were more clearly linked to the Dashboard in Figure 2. For example, 

 * line 178 Describes the 74% of trials are prospectively registered and 

 * line 185 the 38% that have the trial registration number in the abstract. 

These are the first two tiles of the Figure 2 Dashboard, and could be explicitly referred to in the text to help the reader link the text and the dashboard.

2/ I was puzzled that 44% of trials had published within two years but only 7% within five years - which seems contradictory. Can the authors explain this please?

3/ I was expecting and hoping that the Discussion would explore some of the practicalities and barriers for usage in practice by the UMCS. But there seemed to be little on this. Some of the questions I would like to see answered are:

Q1. How much manual work was required to produce the dashboards (that is the number of person-months or costs for doing this) and how much might that be for an annual process?

Q2. Has the dashboard been actively used by any current UMC's (eg the author's institution?) and what has been their reaction to this?

Q3. Clinical trials are important but small proportion of all clinical research, and that proportion will vary widely across UMCs. Might this be included in the Dashboard all the reports to UMC's, and/or are their plans for extending the range of reporting to other types of studies?

Such questions might be usefully addressed in the Discussion.

[LINK]

---

## [Decision Letter · Decision Letter 2]

6 Jan 2023

Dear Dr. Franzen,

Thank you very much for re-submitting your manuscript "Institutional dashboards on clinical trial transparency: A case study for University Medical Centers" (PMEDICINE-D-22-01430R2) for review by PLOS Medicine.

I have discussed the paper with my colleagues and the academic editor and it was also seen again by three reviewers. I am pleased to say that provided the remaining editorial and production issues are dealt with we are planning to accept the paper for publication in the journal.

[LINK]

We look forward to receiving the revised manuscript by Jan 13 2023 11:59PM.   

Sincerely,

Caitlin Moyer, Ph.D.

Senior Editor 

PLOS Medicine

plosmedicine.org

Requests from Editors:

The Academic Editor extends their gratitude for the efforts made in addressing their previous comments. 

Please revise your title to “Institutional dashboards on clinical trial transparency for University Medical Centers: A case study”.

Data Availability Statement: Please confirm the login requirements for the dashboard will be removed prior to publication. 

Please trim your Author Summary such that it contains 2-3 single sentence bullet points under each of the three questions.

Please include key numbers in your Author Summary such as sample size and headline results. 

Figure 3, while suitable for showing the dashboard home page, is not ideal for graphically representing the key findings (the individual panels are too small to appreciate, and axis/data labels are difficult to read). I would suggest providing screenshots of the individual panels in the Supporting Information and citing these in the Results instead. Figure 3 can still be kept in the main body of the manuscript but only cited when referring to the dashboard home page at line 347. 

Comments from Reviewers:

Reviewer #1: I thank authors for their diligent work in revising the manuscript and addressing the reviewers' questions. I do not have any additional comments.

Reviewer #2: The authors have responded to all my comments adequately. One minor comment whiie their response to comment 2 about CT.Gov was fine I wanted to make sure they take care to consider carefully the denominators when comparing platforms.

Reviewer #3: The authors have spent considerable time and effort responding to the suggestions of the editors and the reviewers. This reviewer believes that the authors should be credited with their attention to detail, their own transparency in the process, and the general management of the submission.

[LINK]

---

## [Editor Report · Decision Letter 3]

18 Jan 2023

Dear Dr Franzen, 

On behalf of my colleagues and the Academic Editor, Dr Florian Naudet, I am pleased to inform you that we have agreed to publish your manuscript "Institutional dashboards on clinical trial transparency for University Medical Centers: A case study" (PMEDICINE-D-22-01430R3) in PLOS Medicine.

PRESS

Sincerely, 

Callam Davidson 

Associate Editor 

PLOS Medicine